# Clinical Importance of Regimens in Hepatic Arterial Infusion Chemotherapy for Advanced Hepatocellular Carcinoma with Macrovascular Invasion

**DOI:** 10.3390/cancers13174450

**Published:** 2021-09-03

**Authors:** Takashi Niizeki, Hideki Iwamoto, Tomotake Shirono, Shigeo Shimose, Masahito Nakano, Shusuke Okamura, Yu Noda, Naoki Kamachi, Suzuki Hiroyuki, Miwa Sakai, Ryoko Kuromatsu, Hironori Koga, Takuji Torimura

**Affiliations:** 1Division of Gastroenterology, Department of Medicine, Kurume University School of Medicine, Kurume 830-0011, Japan; niizeki_takashi@kurume-u.ac.jp (T.N.); shirono_tomotake@med.kurume-u.ac.jp (T.S.); shimose_shigeo@med.kurume-u.ac.jp (S.S.); nakano_masahito@med.kurume-u.ac.jp (M.N.); okamura_shyuusuke@kurume-u.ac.jp (S.O.); noda_yuu@med.kurume-u.ac.jp (Y.N.); kamachi_naoki@med.kurume-u.ac.jp (N.K.); suzuki_hiroyuki@med.kurume-u.ac.jp (S.H.); sakai_miwa@med.kurume-u.ac.jp (M.S.); ryoko@med.kurume-u.ac.jp (R.K.); hirokoga@med.kurume-u.ac.jp (H.K.); tori@med.kurume-u.ac.jp (T.T.); 2Iwamoto Internal Medicine Clinic, Kitakyusyu 802-0832, Japan

**Keywords:** hepatocellular carcinoma, vascular invasion, hepatic arterial infusion chemotherapy, macrovascular invasion, low dose FP, New FP

## Abstract

**Simple Summary:**

Although various molecular targeted agents have been approved, the therapeutic outcomes in hepatocellular carcinoma (HCC) with macrovascular invasion (MVI) are still unsatisfactory. Locoregional treatment using hepatic arterial infusion chemotherapy is a promising treatment for MVI-HCC. In the study, we aimed to compare the therapeutic effects of low-dose cisplatin plus 5-fluorouracil (LFP), a conventional HAIC regimen, and New FP (a fine-powder cisplatin suspended with lipiodol plus 5-fluorouracil) for MVI-HCC with preserved liver function. New FP was significantly superior to LFP in all therapeutic outcomes. New FP is a recommended HAIC regimen for the treatment of patients with MVI-HCC.

**Abstract:**

Macroscopic vascular invasion (MVI) is a poor prognostic factor in hepatocellular carcinoma (HCC). Hepatic arterial infusion chemotherapy (HAIC) is a promising treatment in MVI-HCC. However, it is not clear which regimens are suitable for HAIC. In this study, we aimed to compare the therapeutic effects between New FP (a fine-powder cisplatin suspended with lipiodol plus 5-fluorouracil) and low dose FP (LFP/cisplatin plus 5-fluorouracil) in the treatment of MVI-HCC patients with Child–Pugh class A. New FP is a regimen that consists of a fine-powder cisplatin suspended with lipiodol and 5-fluorouracil. Fifty-one patients were treated with LFP, and 99 patients were New FP. We compared the therapeutic effects of LFP and New FP and assessed factors that associated with the therapeutic effects. The median survival and progression-free survival times of LFP and New FP were 16.1/24.7 and 5.4/8.8 months, respectively (*p* < 0.05, *p* < 0.05). The complete response (29%) and objective response rate (76%) of New FP were significantly higher than those of LFP (*p* < 0.001, *p* < 0.01). Factors associated with better therapeutic response were better ALBI-grade and New FP treatment choice. New FP is a more powerful regimen than LFP in HAIC for MVI-HCC. New FP represents a recommended HAIC regimen for the treatment of patients with MVI-HCC.

## 1. Introduction

Hepatocellular carcinoma (HCC) is the fifth most common cancer and is the third leading cause of cancer-related deaths worldwide [1]. Growing advances in the diagnosis and treatment of HCC have contributed to prolonging the prognosis of patients with HCC. However, the World Health Organization reported that more than 0.7 million patients worldwide still died of HCC in 2018. This is partly because many patients with HCC are often diagnosed at the advanced stage [2]. Macrovascular invasion (MVI) and extra-hepatic spread (EHS) are two of the factors that define ‘advanced’ stages of HCC. MVI into the portal vein, hepatic vein, and bile duct are the worst conditions in HCC, respectively [3]. Portal vein tumor thrombus (PVTT) can lead to hepatic failure and portal hypertension-related complications due to obstruction of the portal vein flow. Hepatic vein tumor thrombus (HVTT) can result in lung metastasis and sudden death due to obstruction of the pulmonary artery. Bile duct invasion can lead to the appearance of jaundice. Many studies have reported that these MVIs are independent poor prognostic factors in HCC, respectively. Currently, molecular-targeted agents (MTAs), including sorafenib, lenvatinib, and atezolizumab plus bevacizumab, are the standard treatments for advanced HCC according to the recommended worldwide treatment guidelines [4,5,6]. Although approved MTAs prolong survival even for patients with advanced HCC with MVI (MVI-HCC), their therapeutic effects are unsatisfactory because the underlying prognosis of patients with advanced MVI-HCC is extremely poor [7,8]. Therefore, further progress is needed to improve the treatment of advanced MVI-HCC.

Various therapeutic modalities have been proposed to treat HCC with MVI, including radiation therapy, hepatic resection, and transarterial radioembolization [9,10]. Hepatic arterial infusion chemotherapy (HAIC) is one of the most frequently reported treatments for MVI-HCC [11,12,13,14,15]. HAIC is a locoregional treatment using interventional radiology, which directly and consecutively delivers anti-cancer drugs into the HCC located on the liver by use of catheter techniques. It is theoretically possible to increase local concentrations of anti-cancer drugs in the liver and reduces systemic adverse events (AEs) caused by systemic effects of anti-cancer drugs. According to previous reports concerning HAIC for HCC, it is thought that stronger local control effects are the most important features of HAIC compared to systemic chemotherapy [16,17]. The main regimens of HAIC include low-dose cisplatin (CDDP) combined with 5-fluorouracil (5-FU), also known as low-dose FP (LFP), interferon in combination with 5-FU (FAIT), and CDDP monotherapy [11,12,15]. Previous reports regarding LFP for MVI-HCC have revealed that the objective response rate (ORR) was 35%, and the overall median survival time (MST) was 10.2 months [11,13]. Therapeutic results of other regimens like FAIT and CDDP monotherapy are also similar or inferior to those of LFP. It has been reported that the therapeutic response is important for prolonging the prognosis of patients with MVI-HCC receiving HAIC treatment [18]. Therefore, further improvement of ORR subsequent to regimens of HAIC is an important issue, which should be addressed to further prolong the survival of patients with MVI-HCC.

We describe a regimen for HAIC named “New FP” for the treatment of advanced HCC [19,20,21,22]. New FP is a regimen that consists of a fine-powder CDDP suspended with lipiodol and 5-FU. We have previously reported the therapeutic outcomes of New FP in a single-center analysis and multi-center analysis [19,20,22]. Moreover, a non-randomized multi-center phase 2 trial of New FP and sorafenib in the treatment of HCC has also been reported [21]. Higher ORR and rate of complete response (CR) is a feature of this regimen. Previous studies have reported that New FP significantly prolonged the survival of patients with advanced MVI-HCC when compared to sorafenib. In this study, our aim was to compare the therapeutic efficacy and safety of New FP and LFP, a conventional regimen of HAIC, in patients with HCC.

## 2. Materials and Methods

### 2.1. Study Design

This was a retrospective cohort study in patients diagnosed with HCC who underwent HAIC at a medical center. We compared the therapeutic effects of LFP and New FP. The detailed regimens are described below. The historical data of 51 consecutive patients who received LFP treatment were collected from 1997 to 2007. Ninety-nine consecutive patients were treated with New FP between 2008 and 2018. The data cut-off for this analysis was 30 August 2019. All patients in this study had MVI without EHS, which were diagnosed by contrast-enhanced computed tomography (CT) or magnetic resonance imaging (MRI). Hepatic function was evaluated using the Child-Pugh (C-P) scoring system. All patients were C-P class A (score 5 or 6). Hepatic function was also assessed using the recently developed albumin-bilirubin (ALBI) score and ALBI grading system [23]. The ALBI score was calculated using the following equation: linear predictor = (log_10_ bilirubin μmol/L × 0.66) + (albumin g/L × −0.085). Bilirubin was recorded in mg/dL and albumin in g/dL, but for the calculation of the linear predictor of the ALBI score, these parameters were transformed in the corresponding units (bilirubin in μmol/L and albumin in g/L). The linear predictor of the ALBI score was categorized into three prognostic groups. Grade 1 (less than −2.60), grade 2 (between −2.60 and −1.39), and grade 3 (above −1.39), with a higher ALBI score being associated with an impaired liver function.

The study protocol was approved by the Ethics Committee of Kurume University School of Medicine (No. 19004). Written informed consent regarding HAIC treatment was obtained from each patient. However, the written informed consent for participating in this retrospective data analysis study was waived given the retrospective nature of the data collection. The study design checklist for Strengthening the Reporting of Observational Studies in Epidemiology (STROBE) for cross-sectional studies was consulted [24].

### 2.2. Catheter Implantation Procedures for HAIC

All HAIC treatments were conducted after the insertion of an implanted catheter (Piolax Medical Devices. Inc., Kanagawa, Japan). An implanted catheter for HAIC was indwelled from the right femoral artery. The indwelling approach for the implanted catheter was selected among the gastroduodenal artery (GDA) coiling method, the peripheral hepatic artery fixation method, and the coaxial method, at the discretion of the clinician [25]. Then, GDA, right gastric artery, posterior superior pancreaticoduodenal artery, and accessory left gastric artery were occluded using metallic coils (Piolax Medical Devices. INC, Kanagawa, Japan) to avoid gastroduodenal ulcers or pancreatitis, which could result from the involuntary distribution of the anti-cancer drugs into these arteries. Finally, the port was subcutaneously implanted into the front femoral region (Sofa Port, Nipro Pharma Corporation, Osaka, Japan).

### 2.3. HAIC Regimens

#### 2.3.1. LFP

The LFP regimen is shown in Figure 1A. All anti-cancer drugs used in the LFP regimen were injected using a mechanical infusion pump. As an inpatient LFP regimen, one course consisted of the daily administration of 10 mg of CDDP for 30 min from day 1 to day 5 (Nippon Kayaku, Tokyo, Japan), followed by 250 mg of 5-FU continuously injected for 3 h from day 1 to day 5 (Kyowa Hakko Kogyo, Tokyo, Japan). Day 6 and 7 were the rest period. In principle, this weekly regimen was repeated two or three times as one cycle of LFP. As an outpatient LFP regimen, 20 mg of CDDP and 250 mg of 5-FU were continuously injected every two weeks. The outpatient or inpatient regimens are chosen according to patient/tumor conditions and the therapeutic response. The inpatient regimen of LFP was administered based on the time course of tumor progression, as required. These regimens were continued until the appearance of severe adverse events (AEs) or tumor progression.

#### 2.3.2. New FP

The regimen of New FP is illustrated in Figure 1B. The fine-powder formulation of CDDP was used in the New FP regimen (DDP-H, IA-Call, Nippon Kayaku, Tokyo, Japan). As the inpatient regimen of New FP, 50 mg of fine-powder CDDP was suspended in 5–10 mL of lipiodol, the amount of which was decided by the tumor volume. At day 1, the CDDP-lipiodol suspension was injected from the implanted catheter under angiography and was followed by the injection of 250 mg of 5-FU. Subsequently, 1250 mg of 5-FU was continuously injected using an infusion balloon pump (Surefuser pump, Nipro Pharma Corporation, Osaka, Japan). This regimen was applied once weekly during the first two weeks. As an outpatient regimen of New FP, 20 to 30 mg of CDDP-lipiodol suspension was injected under angiography and was followed by 1250 mg of 5-FU, which was injected using an infusion balloon pump every two weeks. The outpatient or inpatient regimens were chosen according to patient/tumor conditions and the therapeutic response. The inpatient regimen of New FP was administered based on the time course of tumor progression, as required. These regimens were continued until the appearance of severe AEs or tumor progression.

### 2.4. Assessment of Therapeutic Effects and Safety

As the first follow-up, dynamic CT or MRI was performed 1 month after the initial treatment. After the first follow-up, regular imaging examination was conducted every 2-3 months. Scanned images were read and diagnosed by two independent radiologists and one hepatologist. The therapeutic response was assessed according to the modified Response Evaluation Criteria in Solid Tumors (mRECIST) criteria [26]. mRECIST classified therapeutic results into complete response (CR), partial response (PR), stable disease (SD), and progression disease (PD), respectively. The objective response rate (ORR) was defined as CR plus PR. The disease control rate (DCR) was defined as ORR plus SD. The ORR and DCR in the maximal effects were assessed. The progression-free survival (PFS) was defined as the period from the start of the HAIC treatment to the date when PD was detected. The overall survival (OS) was defined as the period from the start of the HAIC treatment to the date when the patient passed away. Post progression survival (PPS) was defined as the period from the date when PD was detected to the date when the patient passed away. The AEs were assessed according to the National Cancer Institute’s Common Terminology Criteria for Adverse Events (CTCAE) version 4.0. We defined grade ≥3 AEs as severe AEs.

### 2.5. Statistical Analysis

All statistical analyses were carried out using JMP statistical analysis software (JMP Pro version 14, SAS institution Inc., Cary, NC, USA). OS, PFS, and PPS were calculated using the Kaplan–Meier method and analyzed with the log-rank test. Factors associated with better and poor prognosis and better therapeutic response were evaluated using univariate and multivariate analyses with the Pearson’s chi-square test and Cox proportional hazards model. Variables associated (*p* < 0.15) with better and poor prognosis and better therapeutic response in univariate analysis were entered into the multivariate regression model. A two-tailed *p*-value < 0.05 was considered statistically significant.

## 3. Results

### 3.1. Patient and Tumor Characteristics

The clinicodemographic characteristics of the patients are summarized in Table 1. The median ages of the LFP and New FP groups were 65.6 ± 10.2 and 69.4 ± 9.0 years old, respectively (*p* < 0.05). All patients were C-P class A. C-P scores 5 and 6 were identified in 26 and 25 patients in the LFP group and in 63 and 36 patients in the New FP group, ALBI-grade 1 and 2 were identified in 11 and 40 patients in the LFP group and in 33 and 66 patients in the New FP group, respectively. All patients in this study had MVI without EHS. PVTT, HVTT, and BDTT were 44, 10, and 1 in the LFP group and 86, 16, and 9 in the New FP group, respectively. In terms of the degree of PVTT, invasion into the second branch, first branch, and trunk of the PV were 13, 17, and 14 in the LFP group and 43, 27, and 16 in the New FP group. The median tumor size in the LFP and New FP groups was 88.0 ± 40.9 and 77.3 ± 39.1 mm, respectively. The median alpha-fetoprotein levels were 1028 (2.6–2,397,149), 957 (1.4–1,977,833) ng/mL. The median des-gamma-carboxy prothrombin (DCP) levels were 3826 (27–129,000) and 1648 (14–2,009,874) mAU/mL (*p* < 0.01). There were no significant differences in patient or tumor characteristics between the LFP and New FP groups except for age (*p* < 0.05) and DCP (*p* < 0.01).

### 3.2. Assessment of the Therapeutic Outcomes in LFP and New FP

Therapeutic outcomes of LFP or New FP are shown in Figure 2. Figure 2A shows the OS curve for all patients in this study (*n* = 150). The median survival time (MST) of all patients was 21.8 months. Figure 2B shows the OS curve of the patients treated with each regimen (Figure 2B). The MST was 16.1 months and 24.7 months in LFP and New FP groups, respectively. There was a significant difference between the two groups (*p* < 0.05, Figure 2B). Figure 2C shows the PFS curve of the patients treated with each regimen (Figure 2C). The PFS was 5.4 months and 8.8 months in LFP and New FP groups, respectively. There was a significant difference between the two groups (*p* < 0.05). Figure 2D shows the PPS curve of the patients treated with each regimen (Figure 2D). The PPS was 6.8 months and 11.5 months in LFP and New FP groups, respectively. There was a trend of longer PPS in New FP, but no significant difference between the two groups (*p* = 0.411). With respect to the therapeutic effects, CR, PR, SD, and PD in LFP were 4%, 43%, 27%, and 26%, respectively (Figure 2E). The CR rate, ORR, and DCR in the LFP group were 4%, 47%, and 74%, respectively, and in the New FP group, the CR, PR, SD, and PD were 29%, 47%, 12%, and 12%, respectively (Figure 2F). The CR rate, ORR rate, and DCR in the New FP group were 29%, 76%, and 88%, respectively. There were significant differences in CR rate, ORR, and DCR between two groups (*p* < 0.001, *p* < 0.01, and *p* < 0.05, respectively. Figure 2G).

We also compared the therapeutic outcomes between MTA sorafenib and HAICs. We used the data of 28 patients treated with sorafenib and without a past history of treatment using HAIC as a historical control (Appendix A). As shown in Appendix A, when compared with three groups, sorafenib, LFP, and New FP, there were significant differences in OS (*p* < 0.001).

### 3.3. Assessment of Adverse Events in LFP and New FP Groups

Severe AEs associated with LFP or New FP are presented in Table 2. The incidence rate of severe AEs was 26.2% and 25.5% in New FP and LFP groups, respectively. There were no significant differences in the development of severe AEs between the two HAIC regimens. HAIC treatment was discontinued owing to severe AEs in 8% and 2% of patients in the New FP and LFP groups, respectively (not significant). Although there were no significant differences in profiles of AEs between the two HAIC regimens, drug allergy, cholangitis, wound dehiscence, and thrombocytopenia were the specific AEs associated with HAIC treatment.

### 3.4. Factors That Associate with Prognosis in Patients Treated with LFP or New FP

To identify the factors that associate with prognosis in patients treated with LFP or New FP in this study, we performed univariate and multivariate analysis (Table 3). The univariate analysis revealed that age (>70 years old), HBs antigen-positive, C-P score 6, ALBI grade 2, DCP > 1600 mAU/mL, maximum tumor diameter >70 mm, >3 tumors located segments, PVTT into trunk, presence of hepatic vein tumor thrombus, therapeutic regimen LFP, were poor prognostic factors. Among these factors, multivariate analysis showed that ALBI grade 2, the number of tumor-located segments >3, and PVTT into trunk were independent factors that influenced poor prognosis (Table 3). These results suggest that tumor progression and liver function are important in HAIC treatment.

### 3.5. Factors Associating with Prognosis in Patients Treated with Each Regimen

To clarify the differences in the therapeutic effects between LFP and New FP, we assessed the factors that associate with poor prognosis in each regimen (Appendix A). In the LFP group, univariate analysis revealed that sex (male), HBs antigen-positive, HCV antibody negative, platelet count >11 × 10^4^/μL, AFP > 400 ng/mL, DCP > 1600 AU/mL, tumor size >70 mm, and PVTT into trunk were poor prognostic factors. Among these factors, multivariate analysis showed that PVTT into trunk was an independent factor that influenced poor prognosis in LFP (Appendix A). With respect to New FP, sex (male), C-P score 6, DCP > 1600 AU/mL, >3 tumors located in segments, and presence of hepatic vein thrombus were poor prognostic factors in univariate analysis. Among these factors, multivariate analysis showed that >3 tumors located in segments were an independent factor that influenced poor prognosis in the New FP regimen (Appendix A). The PVTT into trunk, which was identified as a poor prognostic factor in LFP, was not a poor prognostic factor in New FP. Therefore, we compared the OS of patients with HCC accompanied with PVTT into the portal trunk with LFP and New FP treatment regimen groups. The MST of the LFP and New FP groups were 9.2 and 18.7 months, respectively. The MST of New FP was significantly superior to that of LFP in patients with HCC accompanied with PVTT into the portal trunk (Figure 3A, *p* < 0.01).

### 3.6. Factors Associating with the Therapeutic Response in HAIC Treatment

We conducted an analysis for factors that would associate with a better therapeutic response in HAIC treatment (Table 4). Univariate analysis revealed that C-P score 5, ALBI-grade 1, DCP < 1600 AU/mL, maximum tumor diameter <70 mm, <3 tumors located in segments, and therapeutic regimen New FP were the factors that associated with a better therapeutic response. Among these factors, multivariate analysis showed that ALBI-grade 1 and therapeutic regimen New FP were the independent factors which associate with better therapeutic response in HAIC treatment, respectively (Table 4). These results suggest that choosing New FP as a HAIC regimen is an important factor in achieving a better therapeutic response in HAIC treatment for patients with MVI-HCC.

### 3.7. Comparing OS of the Responders and Non-Responders in LFP and New FP

The therapeutic response was an important factor in achieving a better prognosis in both regimens. Next, we compared the OS of the responders and non-responders in LFP and New FP. The MST of the responders and non-responders in LFP was 23.2 and 7.2 months, respectively (Figure 3B, *p* < 0.001). The MST of the responders and non-responders in New FP was 33.4 and 7.8 months, respectively (Figure 3C, *p* < 0.001). Then, we compared the OS of each responder receiving LFP and New FP regimens (Figure 3D). Interestingly, there was no significant difference between the LFP and New FP groups in the MST of the responder.

## 4. Discussion

In this study, we compared the therapeutic effects of the conventional HAIC regimen LFP with the novel HAIC regimen New FP and found that New FP significantly prolonged PFS and OS. However, there were no significant differences in PPS between LFP and New FP. In addition, the therapeutic response of New FP was remarkably superior to that of LFP. Objective response by mRECIST has been shown to correlate with OS in two clinical trials, and thus additional data is needed to endorse this approach. In the analysis of prognostic factors, “PVTT into trunk” was defined as a potent negative contributor in the LFP regimen but not in the New FP regimen, suggesting that the latter was able to overcome the weaknesses of the former. About “ALBI grade” and “Tumor spread segments”, suggests that lower tumor burden and better liver function may be useful for identifying HCC patients for whom HAIC is associated with better survival. Regarding positive prognostic factors, “therapeutic response” was identified as a significant factor in both regimens for patients with MVI-HCC. Notably, there was no significant difference in MST of responders between the LFP and New FP groups.

The standard treatment for advanced HCC, including MVI-HCC, is systemic administration of MTAs, which has achieved global evidence-based consensus [3,27,28]. Nonetheless, the therapeutic efficacy of MTAs is not satisfactory in patients with highly progressed tumor conditions such as MVI-HCC. The MST of patients with MVI-HCC treated with sorafenib or lenvatinib is 5.6–8.1 months [29]. HAIC is one of the most often-reported and promising treatment modalities for MVI-HCC; however, its efficacy is not always supported by sufficient objective evidence. Several phase 3 randomized clinical trials of HAIC in combination with sorafenib for HCC have been reported. He et al. reported that HAIC in combination with oxaliplatin, 5-FU, and leucovorin (FOLFOX) plus sorafenib significantly improved OS compared with sorafenib monotherapy in patients with HCC complicated with PVTT [30]. HAIC is considered beneficial for controlling tumor progression in locally advanced cases such as those with MVI-HCC. In the study, the MSTs of patients with MVI-HCC treated with LFP and New FP were 16.1 and 24.7 months, respectively. Although we cannot directly compare the therapeutic outcomes, both HAIC regimens seem to be more effective than sorafenib or lenvatinib. We have previously compared the therapeutic effects between New FP and sorafenib, prospectively. The new FP group showed significant prolonged survival compared with sorafenib [21]. Nevertheless, the HAIC regimen should be carefully selected. Kudo et al. reported combination therapy of LFP with sorafenib in the phase 3 SILIUS trial [31]. In this trial, LFP did not prolong OS of all enrolled HCC cases having various tumor conditions, including those with/without MVI and with/without EHS [31]. A sub-analysis of this trial revealed that LFP was effective for HCC with tumor thrombus in the main trunk of the portal vein. Based on these findings, it can be suggested that the therapeutic effect of LFP alone might not be sufficient to improve the OS of patients with MVI-HCC, although LFP could locally control PVTT. In the era of multi-MTAs, we need to reconsider the role of locoregional treatments using HAIC in the treatment of advanced HCC. Our recent study revealed that New FP should be initially administered for locally progressed tumor conditions without extra-hepatic spread (EHS), and sorafenib was effective for HCC with EHS [22].

The need for improvement of OS prompted researchers to develop the novel HAIC regimen New FP. A prospective registry study compared the therapeutic effects of sorafenib alone and New FP in patients with MVI-HCC [21]. Despite the limitations of not being a randomized, large size clinical trial, the above study showed that the New FP regimen significantly prolonged PFS and OS of patients with MVI-HCC compared with sorafenib alone (New FP: PFS 9.5 months, OS 30.4 months; sorafenib: PFS 5.1 months, OS 13.2 months) [19,20]. These findings suggest that New FP is highly effective when compared with systemic MTA therapy in the control of MVI-HCC. However, the therapeutic potential of HAIC regimens depends on their protocols; few studies have compared the therapeutic potential of different HAIC regimens. Kawaoka et al. compared the therapeutic effects of LFP and CDDP monotherapy as HAIC regimens [32] and showed that LFP was superior to CDDP monotherapy. To the best of our knowledge, ours is the first study to compare the therapeutic effects of the conventional HAIC regimen LFP and the novel HAIC regimen New FP. New FP was found to be superior to LFP in all the evaluated items, including OS, PFS, ORR, and CR rate. In the New FP treatment, the suspension of a fine powder of CDDP in lipiodol is essential. This suspended CDDP-lipiodol formulation is considered to possess three additional beneficial effects. First, lipiodol itself possesses a mild embolic effect on tumor vessels, which induces tumor necrosis [33]. Second, there is an enhanced sustained release of CDDP from CDDP-lipiodol suspension. CDDP is gradually released from the concentrated CDDP-lipiodol suspension locally into the tumor when the New FP intervention is administered [34]. Finally, the time-dependent anti-tumor effect is enhanced by the continuous administration of 5-FU using a balloon pump. The sustained release effects of CDDP enhance the biochemical modulatory effect on 5-FU [35]. Based on these mechanisms, it can be suggested that New FP enhanced the anti-tumor effects compared to LFP. Regarding AEs, there were no significant differences in the incidence rate and profiles between the LFP and New FP groups. The incidence rate of severe AEs and discontinuation rate in the LFP and New FP groups in this study are in agreement with those in other studies [36,37]. Moreover, the incidence rate of severe AEs and discontinuation rate in patients treated with MTAs were higher than those treated with HAIC. In the present study, we assessed the negative prognostic factors associated with LFP and identified “PVTT into trunk” as an indicator of poor response to treatment. Notably, in the analysis of negative prognostic factors associated with New FP, “PVTT into trunk” was not defined as a poor prognostic factor, which supports the notion that New FP was able to overcome the weaknesses of LFP treatment. “Tumor spread” was another poor prognostic factor associated with New FP, which also goes to LFP. These results suggest that tumor characteristics, such as “PVTT into trunk” and “extended tumor spread”, would have a significant impact on the prognosis of patients receiving New FP therapy. This phenomenon may be attributable to the embolic effects of lipiodol on tumor vessels in addition to the direct cellular toxicity of anti-cancer agents (i.e., CDDP and 5-FU) used in New FP; LFP lacks the former effect.

From another standpoint, therapeutic response is a common factor associated with better prognosis in patients treated with either LFP or New FP. The ORR of LFP and New FP was 47% and 76%, respectively. A significant difference in therapeutic response was observed between the two regimens. The ORR of other conventional regimens, FAIT and CDDP monotherapy, has been reported to be around 30% and 20%, respectively [18,38]. Thus, the use of conventional HAIC regimens is ceased because of lower ORR. Another study also emphasized the importance of ORR in HAIC treatment for HCC [18,39]. Kudo et al. showed that the OS of responders receiving LFP plus sorafenib was significantly better than that of non-responders [39]. Obi et al. reported that responders to FAIT treatment showed a significantly prolonged survival compared with non-responders [18]. The data shown in our study provide additional insights into ORR–OS relationship: once any HAIC regimen achieves a better therapeutic response (ORR), similar MSTs could also be obtained [39,40]. Indeed, there was no significant difference in OS between responders to LFP and those to New FP in this study. Based on the high ORR and prolonged OS, the New FP regimen should be selected as the first line HAIC regimen for the treatment of patients with advanced HCC, especially those with MVI.

Although this study reported several important findings regarding HAIC regimens used in the treatment of MVI-HCC, it has several limitations. First, this is a retrospective study with a relatively small sample size. However, the therapeutic effects of New FP observed in this study have a significant impact on the treatment of MVI-HCC. In this study, data relative to LFP were collected before the approval of MTAs, indicating that there is a possibility that the post-progression survival was prolonged by MTAs in the New FP group. Of course, MTAs should have contributed to the prolongation of patients’ survival. However, there was no significant difference in PPS between LFP and New FP, which suggested that prolongation of PFS due to New FP directly contributed to that of OS in the study. This temporal bias is an issue resulting from the use of historical control data. In our study, New FP was superior to LFP in terms of OS, PFS, and ORR, suggesting that the HAIC regimen itself was effective regardless of the development of MTAs. Finally, we assessed the therapeutic effects of LFP and New FP only in patients with MVI with preserved liver function. LFP might be an alternative choice for such as multiple tumors without MVI or patients with C-P class B.

## 5. Conclusions

In conclusion, New FP is a more powerful HAIC regimen than LFP for MVI-HCC and shows prolonged PFS, OS, and improved therapeutic response. New FP should be recommended prior to other conventional HAIC regimens. However, it is necessary to provide additional clinical evidence to support the reasonable application of New FP in the treatment of MVI-HCC.

## Figures and Tables

**Figure 1 cancers-13-04450-f001:**
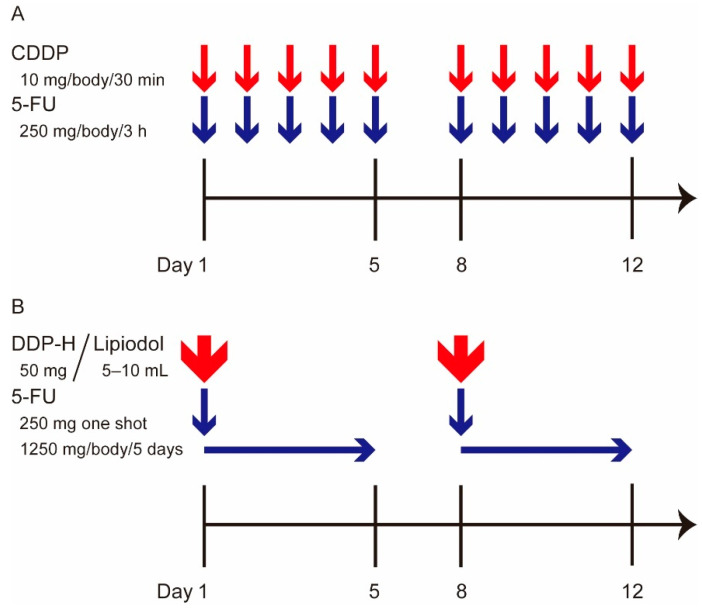
LFP and New FP for inpatient treatment regimens. (**A**) The LFP regimen. As an inpatient regimen, one course consisted of the daily administration of 10 mg of CDDP for 30 min from day 1 to day 5, followed by 250 mg of 5-FU continuously injected for 3 h from day 1 to day 5. Day 6 and 7 was the rest period. In principle, this weekly regimen was repeated 2 or 3 times as one cycle of LFP. (**B**) The New FP regimen. The fine-powder formulation of CDDP (DPP-H) was used in New FP regimen. The inpatient regimen comprised 50 mg of fine-powder DPP-H suspended in 5–10 mL of lipiodol, the amount of which was determined by the tumor volume. At day 1, the DPP-H-lipiodol suspension was injected from the implanted catheter under the angiography, followed by injection of 250 mg of 5-FU. Next, 1250 mg of 5-FU was continuously injected using an infusion balloon pump. This regimen was applied once a week during the first two weeks. LFP: Low dose CDDP plus 5-FU, CDDP: cisplatin, 5-FU: 5-fluorouracil.

**Figure 2 cancers-13-04450-f002:**
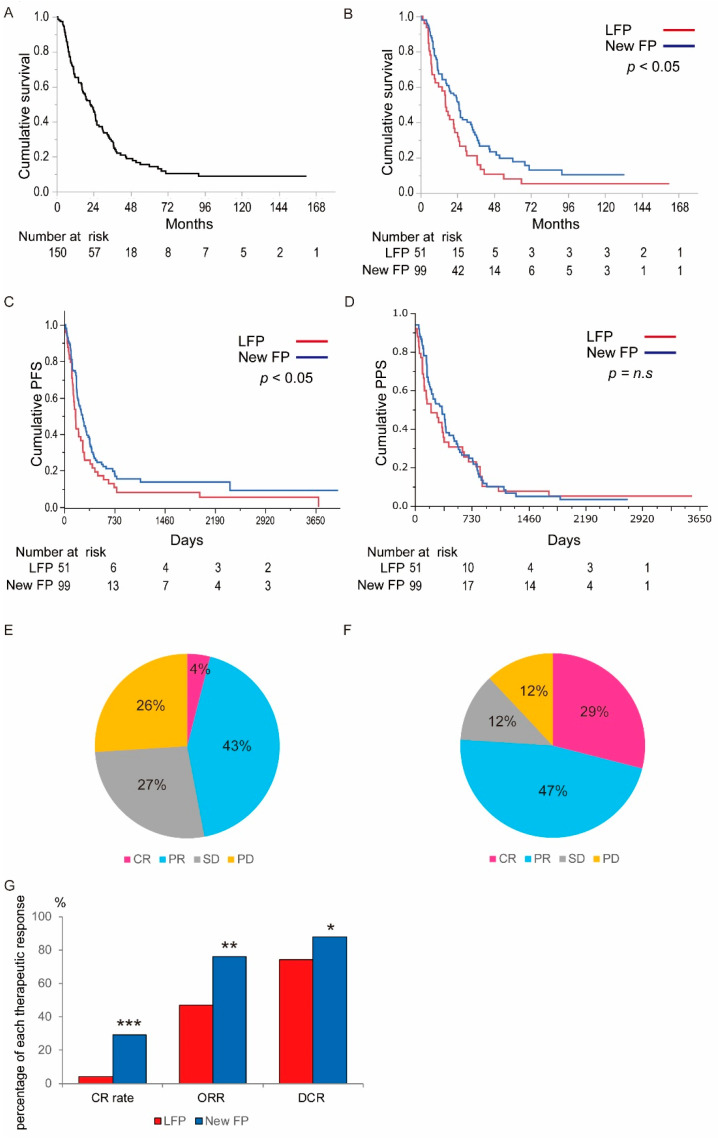
The therapeutic outcomes of LFP and New FP. (**A**) Overall survival curve of the 150 patients treated with either LFP or New FP. The MST of 150 patients was 21.8 months. (**B**) Comparison of the survival curve between LFP and New FP group. The MST of LFP group was 16.1 months. The MST of New FP was 24.7 months. There was a significant difference between two groups (*p* < 0.05). (**C**) Comparison of Progression-free survival between LFP and New FP group. The PFS of LFP group was 5.4 months. The MST of New FP was 8.8 months. There was a significant difference between two groups (*p* < 0.05). (**D**) Comparison of the post-progression survival between LFP and New FP group. The PPS of LFP group was 6.8 months. The PPS of New FP was 11.5 months. There was no significant difference between two groups (*p* = 0.411). (**E**) The therapeutic response of LFP group. In LFP group, CR, PR, SD, and PD were 4, 43, 27, and 26%, respectively. (**F**)The therapeutic response of New FP group. In the New FP group, CR, PR, SD, and PD were 29, 47, 12, and 12%, respectively. (**G**) Comparison of the therapeutic response between LFP and New FP group. The ORR and DCR in LFP group were 47 and 74%, respectively. ORR and DCR in New FP group were 76% and 88%, respectively. There was a significant difference in ORR and DCR between two groups (*p* < 0.001 and *p* < 0.005, respectively.). LFP: Low dose CDDP plus 5-FU, MST: median survival time, CR: complete response, PR: partial response, SD: stable disease, PD: progressive disease, ORR: objective response rate, DCR: disease control rate. * *p* < 0.05, ** *p* < 0.01, *** *p* < 0.001, n.s.: not significant.

**Figure 3 cancers-13-04450-f003:**
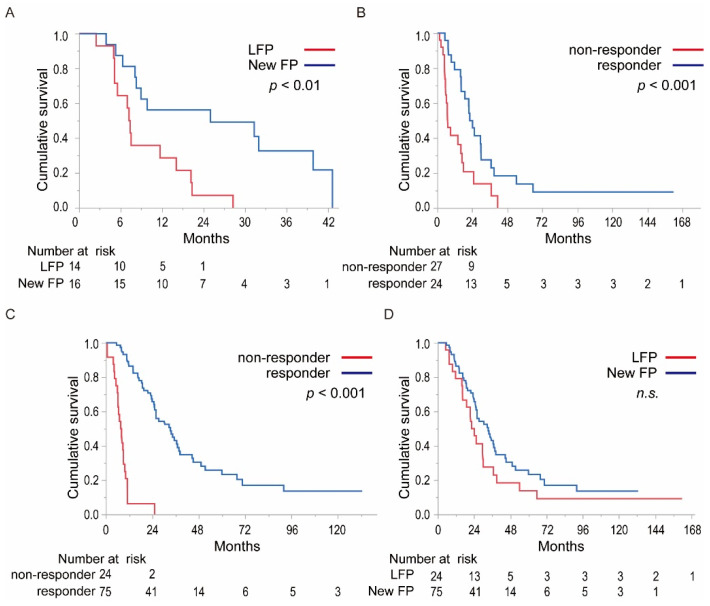
Detailed comparison of the survival curve of patients with HCC treated with either LFP or New FP. (**A**) Comparison of the survival curve of patients accompanied by PVTT into the portal trunk of the LFP and New FP groups. The MST of the LFP and New FP were 9.2 and 18.7 months, respectively. The MST of the New FP was significantly superior to that of LFP in patients with HCC accompanied with PVTT into portal trunk (*p* < 0.01). (**B**) Comparison of the survival curve of the responders and non-responders in the LFP group. The MST of the responders and non-responders in the LFP was 23.2 and 7.2 months, respectively (*p* < 0.001). (**C**) Comparison of the survival curve of the responder and non-responder in the New FP group. The MST of the responders and non-responders in New FP were 33.4 and 7.8 months, respectively (*p* < 0.001). (**D**) Comparison of the survival curve of the responders in the LFP and New FP treatment groups. The MST of the responders to LFP and New FP treatment was 23.2 and 33.4 months, respectively (n.s.). LFP: low dose CDDP plus 5-FU, MST: median survival time, PVTT: portal vein tumor thrombus, n.s.: not significant.

**Table 1 cancers-13-04450-t001:** Patient and tumor characteristics in this study.

Items	Total (150)	New FP (*n* = 99)	Low Dose FP (*n* = 51)	*p*-Value
Age	68.1 ± 9.6	69.4 ± 9.0	65.6 ± 10.2	0.035
Sex	30/120	23/76	7/44	0.158
Etiology (HBV */HCV **/ NBNC ***/HBV + HCV)	27/78/43/2	15/48/35/1	12/30/8/1	0.079
Child-Pugh score (5/6)	89/61	63/36	26/25	0.136
ALBI grade (1/2)	44/106	33/66	11/40	0.134
WBC ^†^ (/μL)	4998 ± 1676	4993 ± 1702	4993 ± 1640	0.926
Plt ^†^^†^ (×10^4^/μL)	15.2 ± 6.8	16.0 ± 7.0	13.7 ± 6.1	0.057
AFP ^†††^ (median) (ng/mL)	992 (1.4–2,397,149)	957 (1.4–1,977,833)	1028 (2.6–2,397,149)	0.310
DCP ^‡^ (median) (mAU/mL)	16,001(14–2,009,874)	1648 (14–2,009,874)	3826 (27–129,000)	0.009
Maximum tumor diameter (mm)	81.0 ± 40.0	77.3 ± 39.1	88.0 ± 40.9	0.094
Tumor number 1/2/3/4/≥5	5/27/19/9/90	4/20/17/5/53	1/7/2/4/37	0.080
Tumor occupancy (<3/≥3)	71/79	49/50	23/28	0.526
PVTT ^‡‡^ (2nd branch/1st Branch/trunk)	56/44/30	43/27/16	13/17/14	0.140
PVTT ^‡‡^ (trunk/1st or 2nd branch)	30/100	16/70	14/30	0.194
HVTT ^‡‡‡^ (present/absent)	26/124	16/83	10/41	0.275
BDTT ^§^ (present/absent)	10/140	9/90	1/50	0.097

* Hepatitis B virus, ** Hepatitis C virus, *** non HBV + non HCV, ^†^ White blood cell count, ^††^ Platelet count, ^†††^ Alphaphetoprotein, ^‡^ Des-gamma-Carboxy Prothrombin, ^‡‡^ Portal vein tumor thrombosis, ^‡‡‡^ Hepatic vein tumor thrombosis, ^§^ Bile duct tumor thrombosis.

**Table 2 cancers-13-04450-t002:** Incidence rate and profiles of adverse events in the New FP and Low-dose FP groups.

Adverse Events	New FP (*n* = 99) (Severe/Discontinuation)	Low-Dose FP (*n* = 51) (Severe/Discontinuation)	*p*-Value
Drug allergy	2/0	0/0	n.s/n.s
Liver failure	2/2	0/0	n.s/n.s
Cholangitis	6/5	1/1	n.s/n.s
Wound dehiscence	0/0	2/0	n.s/n.s
Aneurysm	1/0	0/0	n.s/n.s
Acute kidney disorder	1/0	0/0	n.s/n.s
Biloma	0/0	0/0	n.s/n.s
Thrombocytopenia	8/0	7/0	n.s/n.s
Neutropenia	1/0	3/0	n.s/n.s
Pneumonopathy	2/0	0/0	n.s/n.s
Hepatic infarction	1/0	0/0	n.s/n.s
Somatasthenia	1/1	0/0	n.s/n.s
Abdominal pain	1/0	0/0	n.s/n.s
Total	26/8	13/1	n.s/n.s

n.s.: not significant.

**Table 3 cancers-13-04450-t003:** Univariate and multivariate analysis of the factors associate with poor survival of the patients treated with either LFP or New FP (*n* = 150).

Factors	Univariate Analysis *p*-Value	Multivariate Analysis *p*-Value, Hazard Ratio (95% CI)
Age (≥70 years)	0.044	0.440
Sex (Male)	0.784	-
HBs * Ag (+)	0.055	0.177
HCV ** Ab (−)	0.271	-
Child-Pugh score 6	0.016	0.413
ALBI-grade 1	0.005	0.012, 1.912 (0.058–0.484)
Platelet count (≥11 × 10^4^/μL)	0.625	-
AFP *** (≥400 ng/mL)	0.784	-
DCP ^†^ (≥1600 mAU/mL)	0.003	0.126
Maximum tumor diameter (≥70 mm)	0.044	0.483
Tumor located segments (≥3)	0.003	0.026, 1.58 (−0.357 to −0.022)
Grade of PVTT ^††^ (trunk)	0.0004	0.019, 1.71 (−0.460 to −0.053)
Hepatic vein tumor thrombus (+)	0.012	0.086
Bile duct tumor thrombus (+)	0.681	-
Regimen (LFP ^†††^)	0.029	0.129

* Hepatitis B virus, ** Hepatitis C virus, *** Alphaphetoprotein, ^†^ Des-gamma-Carboxy Prothrombin, ^††^ Portal vein tumor thrombus, ^†††^ Low dose CDDP plus 5-FU.

**Table 4 cancers-13-04450-t004:** Univariate and multivariate analysis of the factors associate with the better therapeutic effects of either LFP or New FP (*n* = 150).

Factors	CR or PR (*n* = 99)	SD or PD (*n* = 51)	Univariate Analysis *p*-Value	Multivariate Analysis *p*-Value, (Logarithmic Value)
Age (<70 years)	52/47	23/28	0.388	-
Sex (Female)	76/23	44/7	0.158	-
HBs * Ag (−)	16/83	13/38	0.159	-
HCV ** Ab (+)	58/41	22/29	0.092	0.54
ALBI-grade (1)	36/63	8/43	0.006	0.033, (2.17)
Platelet count (>11 × 10^4^/μL)	68/31	41/10	0.121	0.53
AFP *** (<400 ng/mL)	56/43	32/19	0.465	-
DCP ^†^ (<1600 mAU/mL)	46/53	39/12	0.0003	0.37
Maximum tumor diameter (<70 mm)	38/61	35/16	0.0004	0.26
Tumor located segments (<3)	45/54	17/34	0.0130	0.17
Grade of PVTT ^††^ (2nd or 1st branch)	18/81	12/39	0.442	0.38
Hepatic vein tumor thrombus (−)	14/85	12/39	0.157	-
Regimen (New FP ^†††^)	75/24	24/27	0.0005	0.0008, (3.12)

* Hepatitis B virus, ** Hepatitis C virus, *** Alphaphetoprotein, ^†^ Des-gamma-Carboxy Prothrombin, ^††^ Portal vein tumor thrombus, ^†††^ DDP-H suspended with lipiodol plus 5-FU.

## Data Availability

Data is contained within the article or Appendix A.

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
