# Peer review of "Clinical Importance of Regimens in Hepatic Arterial Infusion Chemotherapy for Advanced Hepatocellular Carcinoma with Macrovascular Invasion"

_cancers, 2021, doi:10.3390/cancers13174450_

Round 1

Reviewer 1 Report

Authors seek to compare the therapeutic effects of LFP and New FP HAIC regimen for the treatment of patients with MVI-Child A-HCC.

The Authors provide a nice and concise description of the state of the art in the use of HAIC for HCC. The background, M&M and results are well described and the Figures and the Tables provide the necessary support.

Minor cons:

TITLE: I suggest to change it to a shorter form.

Line 16 and 24: Is better to declare FP at first use; e.g.: fine-powder cisplatin and 5-fluorouracil (New-FP)

Line 78: typo in the quotation mark

Line 80-81: the sentence has no logical connection and is not concluded. To be verified.

Line 125: can you please specify why/when you choose inpatient or outpatient treatment? According to?

Line 135: AE must be declared

Figure 2: error in the numbering (two E, G missing)

Line 326: typo

Reviewer 2 Report

Niizeki et al. reported that new FP, a regimen of hepatic arterial infusion chemotherapy,  is superior to low dose FP in treatment of advanced HCC with macroscopic vascular invasion.

  1. Now the standard regimen for the advanced HCC with macroscopic vascular invasion are molecular targeted therapies such as sorafenib, lenvatinib, etc. Authors should use these treatment results as control and discuss more.
  2. Do not use “FP” when they first appeared in the title, abstract and introduction. Authors should explain them.
  3. In page 3, “Study protocol was approved by the Ethics Committee of the University (No.19004). Written informed consent regarding HAIC treatment was obtained from each patient….” Which University is it?
  4. Study protocol was approved by the Ethics Committee of the University (No.19004). 107 Written informed consent regarding HAIC treatment was obtained from each patient.
  5. In the sorafenib-era, how do the authors select patients for this treatment? 

Round 2

Reviewer 2 Report

Authors should include patients treated with sorafenib as control.

Author Response

【Answer】

Thank you very much for your important suggestion. We added another analysis compared with sorafenib and HAICs. There were significant differences between sorafenib and HAICs in survival. We added these important data in supplementary Figure 1 and supplementary Table 1. We also added the descriptions about them in line 234-238.